# AEVA: Black-box Backdoor Detection Using Adversarial Extreme Value Analysis

**Junfeng Guo, Ang Li, Cong Liu**
Department of Computer Science
The University of Texas at Dallas
{jxg170016,angli,cong}@utdallas.edu

## Abstract

Deep neural networks (DNNs) are proved to be vulnerable against backdoor attacks. A backdoor is often embedded in the target DNNs through injecting a *backdoor trigger* into training examples, which can cause the target DNNs misclassify an input attached with the backdoor trigger. Existing backdoor detection methods often require the access to the original poisoned training data, the parameters of the target DNNs, or the predictive confidence for each given input, which are impractical in many real-world applications, e.g., on-device deployed DNNs. We address the black-box hard-label backdoor detection problem where the DNN is fully black-box and only its final output label is accessible. We approach this problem from the optimization perspective and show that the objective of backdoor detection is bounded by an adversarial objective. Further theoretical and empirical studies reveal that this adversarial objective leads to a solution with highly skewed distribution; a singularity is often observed in the adversarial map of a backdoor-infected example, which we call the *adversarial singularity phenomenon*. Based on this observation, we propose the *adversarial extreme value analysis* (AEVA) to detect backdoors in black-box neural networks. AEVA is based on an extreme value analysis of the adversarial map, computed from the monte-carlo gradient estimation. Evidenced by extensive experiments across multiple popular tasks and backdoor attacks, our approach is shown effective in detecting backdoor attacks under the black-box hard-label scenarios.

## 1 Introduction

Deep Neural Networks (DNNs) have pervasively been used in a wide range of applications such as facial recognition (Masi et al., 2018), object detection (Szegedy et al., 2013; Li et al., 2022), autonomous driving (Okuyama et al., 2018), and home assistants (Singh et al., 2020; Zhai et al., 2021). In the meanwhile, DNNs become increasingly complex. Training state-of-the-art models requires enormous data and expensive computation. To address this problem, vendors and developers start to provide pre-trained DNN models. Similar to softwares shared on GitHub, pre-trained DNN models are being published and shared on online venues like the BigML model market, ONNX zoo and Caffe model zoo.

Since the dawn of software distribution, there has been an ongoing war between publishers sneaking malicious code and backdoors in their software and security personnel detecting them. Recent studies show that DNN models can contain similar backdoors, which are induced due to contaminated training data. Sometimes models containing backdoors can perform better than the regular models under untampered test inputs. However, under inputs tampered with a specific pattern (called trojan trigger) models containing backdoors can suffer from significant accuracy loss.

There has been a significant amount of recent work on detecting the backdoor triggers. However, those solutions require access to the original poisoned training data (Chen et al., 2019a; Tran et al., 2018; Huang et al., 2022), the parameters of the trained model (Chen et al., 2019b; Guo et al., 2020; Liu et al., 2019; Wang et al., 2019; 2020; Dong et al., 2021; Kolouri et al., 2020), or the predicted confidence score of each class (Dong et al., 2021). Unfortunately, it is costly and often impractical for the defender to access the original poisoned training dataset. In situations when DNNs are

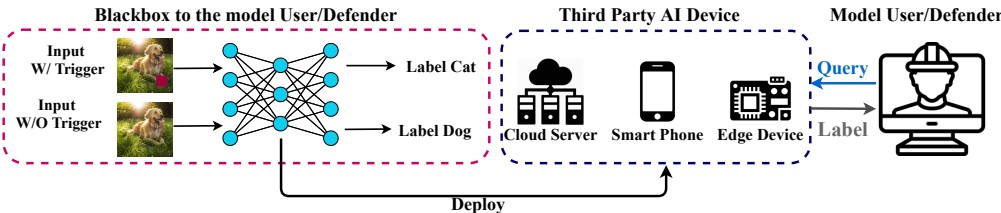

Figure 1: An illustration of the black-box hard-label backdoors.

deployed in safety-critical platforms (Fowers et al., 2018; Okuyama et al., 2018; Li et al., 2021b) or in cloud services (Fowers et al., 2018; Chen et al., 2019c), it is also impractical to access either their parameters or the predicted confidence score of each class (Chen et al., 2019c; 2020b).

We present the *black-box hard-label backdoor detection* problem where the DNN is a fully black-box and only its final output label is accessible (Fig. 1). Detecting backdoor-infected DNNs in such black-box setting becomes critical due to the emerging model deployment in embedded devices and remote cloud servers. In this setting, the typical optimization objective of backdoor detection (Chen et al., 2019b; Liu et al., 2019) becomes impossible to solve due to the limited information. However, according to a theoretical analysis, we show that the backdoor objective is bounded by an adversarial objective, which can be optimized using Monte Carlo gradient estimation in our black-box hard-label setup. Further theoretical and empirical studies reveal that this adversarial objective leads to a solution with highly skewed distribution; a singularity is likely to be observed in the adversarial map of a backdoor-infected example, which we call the *adversarial singularity phenomenon*.

Based on these findings, we propose the *adversarial extreme value analysis* (AEVA) algorithm to detect backdoors in black-box neural networks. The AEVA algorithm is based on an extreme value analysis (Leadbetter, 1991) on the adversarial map. We detect the adversarial singularity phenomenon by looking at the *adversarial peak*, *i.e.*, maximum value of the computed adversarial perturbation map. We perform a statistical study which reveals that, there is around $60\%$ chance that the adversarial peak of a random sample for a backdoor-infected DNN is larger than the adversarial peak of any examples for an uninfected DNN. Inspired by the Univariate theory, we propose a global adversarial peak (GAP) value by sampling multiple examples and choosing the maximum over their adversarial peaks, to ensure a high success rate. Following previous works (Dong et al., 2021; Wang et al., 2019), the Median Absolute Deviation (MAD) algorithm is implemented on top of the GAP values to test whether a DNN is backdoor-infected.

Through extensive experiments across three popular tasks and state-of-the-art backdoor techniques, AEVA is proved to be effective in detecting backdoor attacks under the black-box hard-label scenarios. The results show that AEVA can efficiently detect backdoor-infected DNNs, yielding an overall detection accuracy $\geq 86.7\%$ across various tasks, DNN models and triggers. Rather interestingly, when comparing to two state-of-art white-box backdoor detection methods, AEVA yields comparable performance, even though AEVA is a black-box method with limited access to information.

We summarize our contributions as below [1]:

1. To the best of our knowledge, we are the first to present the black-box hard-label backdoor detection problem and provide an effective solution to this problem.

2. We provide a theoretical analysis which shows backdoor detection optimization is bounded by an adversarial objective. And we further reveal the adversarial singularity phenomenon where the adversarial perturbation computed from a backdoor-infected neural network is likely to suffer from a highly skewed distribution.

3. We propose a generic backdoor detection framework AEVA, optimizing the adversarial objective and performing extreme value analysis on the optimized adversarial map. AEVA is applicable to the black-box hard-label setting with a Monte Carlo gradient estimation.

---

[1]Code:https://github.com/JunfengGo/AEVA-Blackbox-Backdoor-Detection-main

    4. We evaluate AEVA on three widely-adopted tasks with different backdoor trigger implementations and complex black-box attack variants. All results suggest that AEVA is effective in black-box hard-label backdoor detection.

## 1.1 RELATED WORK

**Backdoor attacks.** BadNets (Gu et al., 2019) is probably the first work on backdoor attacks against DNNs, which causes target misclassification to DNNs through injecting a small trigger into some training samples and mis-labeling these samples with a specific target class. Trojaning attack (Liu et al., 2018) generates a trigger which can cause the maximum activation value of certain selected neurons with limited training data. Chen *et al.* (Chen et al., 2017) further propose to perform backdoor attacks under a rather weak threat model where the attacker cannot access the target model and training dataset. Most recently, a set of backdoor attacks (Ji et al., 2018; Liu et al., 2020; Saha et al., 2020; Turner et al., 2019; Yao et al., 2019) have been proposed, which are built upon existing work but focused on various specific scenarios, *e.g.*, physical-world, transfer learning, *etc*.

**Backdoor detection (white-box).** A few works (Chen et al., 2019a; Tran et al., 2018) are proposed to detect the backdoor samples within the training dataset. Chen et al. (2019a) propose a neuron activation clustering approach to identify backdoor samples through clustering the training data based on the neuron activation of the target DNN. Tran et al. (2018) distinguish backdoor samples from clean samples based on the spectrum of the feature covariance of the target DNN. Few other works focus on detecting backdoor-infected DNNs (Liu et al., 2019; Wang et al., 2019). ABS (Liu et al., 2019) identifies the infected DNNs by seeking compromised neurons representing the features of backdoor triggers. Neural Cleanse (Wang et al., 2019) conducts the reverse engineering to restore the trigger through solving an optimization problem. Recent works (Chen et al., 2019b; Guo et al., 2020; Wang et al., 2020) improve Neural Cleanse with better objective functions. However, these methods are all white-box based, requiring access to model parameters or internal neuron values.

**Backdoor detection (black-box).** A recent work (Chen et al., 2019b) claims to detect backdoor attacks in the "black-box" settings. However, their method still need the DNN's parameters to train a separate generator (Goodfellow et al., 2014). So, strictly speaking, their method is not "black-box", which is also revealed by (Dong et al., 2021). To the best of our knowledge, Dong et al. (2021) is the only existing work on detecting backdoor-infected DNNs in the black-box settings. However, their method requires the predictive confidence score for each input to perform the NES algorithm (Wierstra et al., 2014), which weakens its practicability. Our work differs from all previous methods in that we address a purely black-box setup where only the hard output label is accessible. Both model parameters and training examples are inaccessible in the black-box hard-label setting.

## 2 PRELIMINARIES: BLACK-BOX BACKDOOR ATTACK AND DEFENSE

### 2.1 THREAT MODEL

Our considered threat model contains two parts: the *adversary* and the *defender*. The threat model of the adversary follows previous works (Chen et al., 2017; Gu et al., 2019; Liu et al., 2018). In this model, the attacker can inject an arbitrary amount of backdoor samples into the training dataset and cause target misclassification to a specific label without affecting the model's accuracy on normal examples. From the perspective of the defender, we consider the threat model with the weakest assumption, in which the poisoned training dataset and the parameters of the target DNNs $f(\cdot)$ are inaccessible. Moreover, the defender can only obtain the final predictive label for each input from the target DNNs and a validation set (40 images for each class). Therefore, the defender can only query the target DNNs to obtain its final decisions. The defender's goal is to identify whether the target DNNs is infected and which label is infected.

### 2.2 PROBLEM DEFINITION

Consistent with prior studies (Dong et al., 2021; Kolouri et al., 2020; Wang et al., 2019; Huang et al., 2022), we deem a DNN is backdoor infected if one can make an arbitrary input misclassified as the target label, with minor modification to the input. Without loss of generability, given the original

input $x \in \mathbb{R}^n$, the modified input containing the backdoor trigger can be formulated as:

$$\hat{x} = b(x) = (1 - m) \odot x + m \odot \Delta, \tag{1}$$

where $\Delta \in \mathbb{R}^n$ represents the backdoor trigger and $\odot$ represents the element-wise product. $m \in \{0, 1\}^n$ is a binary mask that ensures the position and magnitude of the backdoor trigger. Typically, $||m||_1$ ought to be very small to ensure the visually-indistinguishability between $\hat{x}$ and $x$.

We define $f(\cdot; \theta)$ as the target DNN with parameters $\theta$, which is a function that maps the input $x$ to a label $y$. $\phi(x, y; \theta) \in [0, 1]$ is defined as the probability that $x$ is classified as $y$ by $f(\cdot; \theta)$. The argument $\theta$ is sometimes omitted in favor of simplicity. For $f$ to mislabel $\hat{x}$ as the target label $y_t$, there are two possible reasons: the backdoored training data either contains the same trigger but mislabeled as $y_t$ or contains similar features as the normal inputs (not belonging to $y_t$, but visually similar to $y_t$) and labeled as $y_t$.

$$\hat{\theta} = \arg\min_{\theta} \sum_{i=1}^{N_o} \ell(\phi(x_i, y_i; \theta), y_i) + \sum_{j=1}^{N_b} \ell(\phi(\hat{x}_j, y_t; \theta), y_t), \tag{2}$$

where there are $N_o$ original training examples and $N_b$ backdoor examples. $\ell(\cdot)$ represents the cross-entropy loss function; $x_i$ and $y_i$ represent the training sample and its corresponding ground-truth label, respectively. In the inference phase, the target DNN will predict each $\hat{x}$ as the target label $y_t$, which can be formulated as: $f(\hat{x}) = y_t$. Prior works (Dong et al., 2021; Wang et al., 2019; Chen et al., 2019b) seek to reveal the existence of a backdoor through investigating whether there exists a backdoor trigger with minimal $||m||_1$ that can cause misclassification to a certain label.

We mainly study the sparse mask since it is widely used in previous works (Wang et al., 2020; Dong et al., 2021; Chen et al., 2019b; Guo et al., 2020; Wang et al., 2019; Gu et al., 2019). In cases when the mask becomes dense, the mask $m$ has to contain low magnitude continuous values to ensure the backdoor examples visually indistinguishable. However, when the value of $m$ is small, every normal example $x_i$ becomes near the decision boundary of function $\phi$. It is revealed by Taylor theorem, $\phi(\hat{x}_i) = \phi((1 - m)x_i + m\Delta) = \phi(x_i + m(\Delta - x_i)) \approx \phi(x_i) + m(\Delta - x_i)^\intercal \nabla_x \phi$. And the function output label changes dramatically from $\phi(x_i)$ to $\phi(\hat{x}_i)$, so the gradient $\nabla_x \phi$ has to have a large magnitude, which results in a non-robust model sensitive to subtle input change. Such models become impractical because a small random perturbation on the input can lead to significantly different model output. More details with empirical study can be found in the Appendix A.

## 2.3 BLACK-BOX BACKDOOR DETECTION AND ITS CHALLENGES

Most previous works (Wang et al., 2019; Dong et al., 2021; Chen et al., 2019b) focus on reverse-engineering the backdoor trigger for each target label $y_t$ using the following optimization:

$$\arg\min_{m, \Delta} \sum_i \ell(\phi((1 - m) \odot x_i + m \odot \Delta, y_t), y_t) + \beta ||m||_1, \tag{3}$$

where $\beta$ is the balancing parameter. Unfortunately, solving Eq. 3 is notoriously hard in the black-box hard-label setting since $\theta$ is unknown. Additional difficulty comes with the fact that an effective zero-th order gradient estimation requires each example $x_i$ superimposed with the trigger to be close to the decision boundary of $y_t$ (Chen et al., 2020a; Cheng et al., 2020; Li et al., 2020). However, such condition is hard to achieve in the hard-label blackbox settings.

## 3 ADVERSARIAL EXTREME VALUE ANALYSIS FOR BACKDOOR DETECTION

Our AEVA framework is introduced in this section. We first derive an upper bound to the backdoor detection objective in Eq. 3 and find its connection to adversarial attack. The adversarial singularity phenomenon is further revealed in both theoretical results and empirical studies. Based on our findings, we propose the Global Adversarial Peak (GAP) measure, computed by extreme value analysis on the adversarial perturbation. The GAP score will be finally used with Median Absolute Deviation (MAD) to detect backdoors in neural networks. Monte Carlo gradient estimation is further introduced in order to solve the adversarial objective in the black-box setting.

### 3.1 ADVERSARIAL OBJECTIVE AS AN UPPER BOUND OF THE BACKDOOR OBJECTIVE

We present in this section the connection between the backdoor objective in Eq. 3 with the objective in adversarial attacks. To begin with, we first transform the backdoor masking equation $(1 - m) \odot x_i + m \odot \Delta = x_i + \mu - m \odot x_i$, where $\mu = m \odot \Delta$. Following this, the optimization in Eq. 3 converts to minimizing the following term:

$$F = \frac{1}{N} \sum_{i=1}^{N} \ell(\phi(x_i + \mu - m \odot x_i, y_t; \theta), y_t) . \tag{4}$$

We further introduce an important result from the Multivariate Taylor Theorem below.

**Lemma 1.** *Given a continuous differentiable function $f(\cdot)$ defined on $\mathbb{R}^n$ and vectors $x, h \in \mathbb{R}^n$, for any $M \geq |\partial f / \partial h_i|, \forall 1 \leq i \leq n$, we have $|f(x + h) - f(x)| \leq M \|h\|_1$.*

The lemma can be proved using the Lagrange Remainder. Please refer to the Appendix B.1 for a detailed proof. According to the lemma, we have

$$\left| F - \frac{1}{N} \sum_{i=1}^{N} \ell(\phi(x_i + \mu, y_t; \theta), y_t) \right| \leq \frac{1}{N} \sum_{i=1}^{N} C \|m \odot x_i\|_1 \leq C \|m\|_1 = C \|\mu\|_0 , \tag{5}$$

where the latter inequality holds because each example $x_i \leq 1$ is bounded and the last equality $\|m\|_1 = \|m \odot \Delta\|_0 = \|\mu\|_0$ holds because $m$ is a binary mask. Then, we can see that the adversarial objective is an upper bound of the objective in backdoor detection, *i.e.*, $F \leq \frac{1}{N} \sum_i \ell(\phi(x_i + \mu), y_t) + C \|\mu\|_0$. Instead of optimizing the original Eq. 3, we here propose to minimize the $\ell^0$-regularized objective. While $\ell^0$ enforces the sparsity of the solution, optimizing it is NP-hard. In practice, it is replaced with an $\ell^1$-norm as an envelope of the objective (Ramirez et al., 2013; Donoho, 2006):

$$\hat{\mu} = \arg \min_{\mu} \sum_i \ell(\phi(x_i + \mu, y_t; \theta), y_t) + \lambda \|\mu\|_1 . \tag{6}$$

This adversarial optimization can be solved using Monte Carlo gradient estimation in our black-box setting. More details are described in Section 3.4. In the following, we will first discuss how the solution to this adversarial objective can be used to detect backdoors in neural networks.

### 3.2 THE ADVERSARIAL SINGULARITY PHENOMENON

A natural and important question here is what the adversarial perturbation $\hat{\mu}$ would look like when the input is infected with backdoor attacks. A direct analysis on deep neural networks is hard. So, we start by analyzing a linear model to shed light on the intuition of our approach.

**Lemma 2.** *Given a linear model parameterized by $\theta$ optimized on $N_o$ original training examples and $N_b$ backdoored training examples with an objective in Eq. 2 and a mean-squared-error loss, the adversarial solution $\hat{\mu}$ optimized using Eq. 6 will be dominated by input dimensions corresponding to backdoor mask $m$ when $N_b$ is large, i.e., $\lim_{N_b \to \infty} \|(1 - m) \odot \hat{\mu}\|_1 / \|\hat{\mu}\|_1 = 0$.*

The lemma can be proved by least square solutions and exact gradient computation. Detailed proof can be found in the Appendix B.2. The lemma reveals that the majority of the mass in the adversarial perturbation will be occupied in the mask area. Since the mask $m$ is usually sparse, it is reasonable to expect a highly skewed distribution in the adversarial map $\hat{\mu}$.

We suspect that such skewed distributions might also occur in the adversarial map of a deep neural network. While a thorough analysis on DNN is hard, recent studies in Neural Tangent Kernel (NTK) (Jacot et al., 2018) have shown that a deep neural network with infinite width can be treated as kernel least square. So, we further extended our analysis to a $k$-way kernel least square classifier as the model and a cross-entropy loss used in the adversarial analysis.

**Lemma 3.** *Suppose the training dataset consists of $N_o$ original examples and $N_b$ backdoor examples, i.i.d. sampled from uniform distribution and belonging to $k$ classes. Let $\phi$ be a multivariate kernel regression with the objective in Eq. 2, an RBF kernel. Then the adversarial solution $\hat{\mu}$ to Eq. 6 under cross-entropy loss should satisfy that $\lim_{N_b \to \infty} \mathbb{E}[(1 - m) \odot \hat{\mu}] = 0$.*

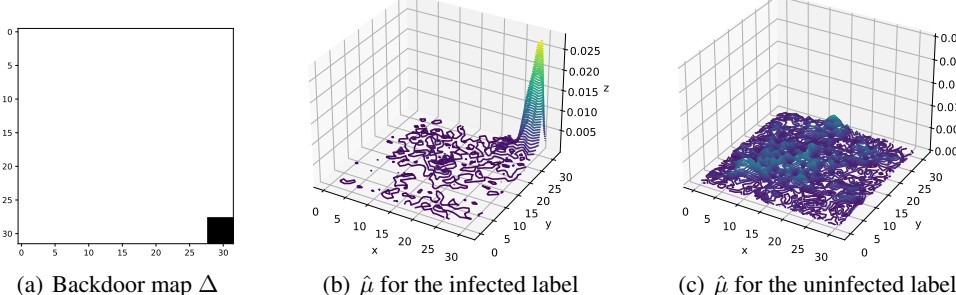

| (a) Backdoor map $\Delta$ | (b) $\hat{\mu}$ for the infected label | (c) $\hat{\mu}$ for the uninfected label |

Figure 2: The distribution of values in the normalized adversarial perturbation $\mu$ for the infected (b) and uninfected (c) labels, using a backdoor map in (a). $x, y$ axes represent the image space; $z$ axis represents the absolute value of the normalized $\mu$ corresponding to each pixel location.

The proof can be found in Appendix B.3. The lemma reveals a similar result as the linear case that the majority mass of the adversarial perturbation (in expectation) stays in the mask area when sufficient backdoor examples are used. Although the lemma does not directly address a deep neural network, we hope this result could help understand the problem better in the NTK regime.

We perform an empirical study on DNNs to validate our intuition. Following previous works (Dong et al., 2021; Wang et al., 2019; 2020), we implement BadNets (Gu et al., 2019) as the backdoor attack method. We randomly select a label as the infected label and train a ResNet-44 on CIFAR-10 embedded with the backdoor attack, resulting in an attack success rate of $99.87\%$. The backdoor trigger is a 4x4 square shown in Fig. 2(a), which occupies at most $1.6\%$ of the whole image. The adversarial perturbations $\hat{\mu}$ (by optimizing Eq. 6) are generated for both infected and uninfected labels. The perturbations are normalized to a heat map as $|\hat{\mu}|/||\hat{\mu}||_1$ and shown in Fig. 2. A high concentration of mass is clearly observed from the infected case in Figure 2(b). We call it *adversarial singularity phenomenon*. This motivates our adversarial extreme value analysis (AEVA) method.

### 3.3 ADVERSARIAL EXTREME VALUE ANALYSIS

A natural way to identify singularity is to look at the peak value of the distribution. We define the *adversarial peak* $\mu_{max} = \max \mu_{ij}$ for an adversarial perturbation $\mu$. To gain a statistical understanding of the relationship between adversarial peak and backdoor attacks, We randomly sample 1000 normalized $\mu$ for the uninfected and infected labels and plot the distribution of $\mu_{max}$ in Fig. 3. The experimental setup is consistent with the empirical study in Sec. 3.2. We notice from the figure that $\mu_{max}$ of the infected labels reaches a much larger range compared to that of uninfected labels. However, they still overlap; around $47\%$ infected $\mu_{max}$ stays within the uninfected range (the area with both blue and orange bars). This result suggests that

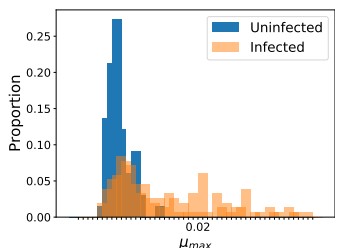

Figure 3: The distributions of adversarial peak $\mu_{max}$ for infected labels and uninfected labels.

*the adversarial singularity phenomenon does not always occur in backdoor-infected DNNs*. It is reasonable because some samples are inherently vulnerable to adversarial attacks and their adversarial perturbations have similar properties as backdoors, which reduces the chance of singularity.

Now, we know if we define a threshold $T$ as the maximum value of the uninfected labels, the probability of a backdoor-infected $\mu_{max}$ being smaller than $T$ is 0.47, *i.e.*, $P(\mu_{max} < T) = 0.47$. Taking one example is impossible to know whether a label is infected. However, according to the Univariate Theory (Gomes & Guillou, 2015), if we sample $k$ examples of $\mu_{max}$ for the same label,

$$P(\max\{\mu_{max}^1, \mu_{max}^2, \ldots, \mu_{max}^k\} < T) = (P(\mu_{max} < T))^k . \tag{7}$$

Here we name the maximum value over all the $k$ adversarial peaks as the *Global Adversarial Peak* (GAP). We vary the choice of $k$ in Figure 4 and we can see that the chance of GAP value being lower than $T$ is diminishing. For example, if we take $k = 6$, then $P(GAP < T) = 0.47^6 = 0.01$ and the success rate of identifying a backdoor-infected label is $99\%$. Please be advised that, given

the long-tail distribution of the infected $\mu_{max}$, this threshold $T$ is not sensitive and can be made even larger to be safe to include all uninfected labels. However, for larger $T$ values, we need a larger $k$ to ensure a desired success rate.

### 3.4 BLACK-BOX OPTIMIZATION VIA GRADIENT ESTIMATION

After investigating the unique properties of backdoor-infected DNNs in the white-box settings, we here illustrate how to exploit such properties under a rather restricted scenario as we consider. In the white-box settings, the free access to $\theta$ can enable us to compute the $\nabla_x \phi(x, y_t)$ accurately, resulting in $\mu$ with minimum $||\mu||_1$ through optimizing Eq. 6. Regarding the black-box settings, we propose to leverage the zero-order gradient estimation technique (Fu & Hu, 2012) to address the challenge for calculating $\nabla_x \phi(x, y_t)$. We choose Monte Carlo based method (Fu & Hu, 2012) for obtaining the estimated gradient $\widetilde{\nabla}\phi_x(x, y_t)$, which sends several inputs and utilizes the corresponding outputs to estimate the gradient. More details about gradient estimation can be found in Appendix C.

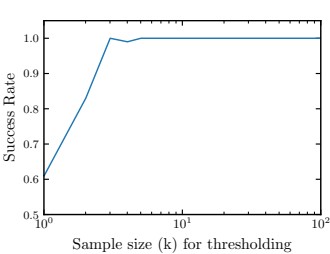

Figure 4: The proportion of infected labels whose $\max\{\mu_{max}^i\}$ larger than that of uninfected ones.

### 3.5 FINAL ALGORITHM

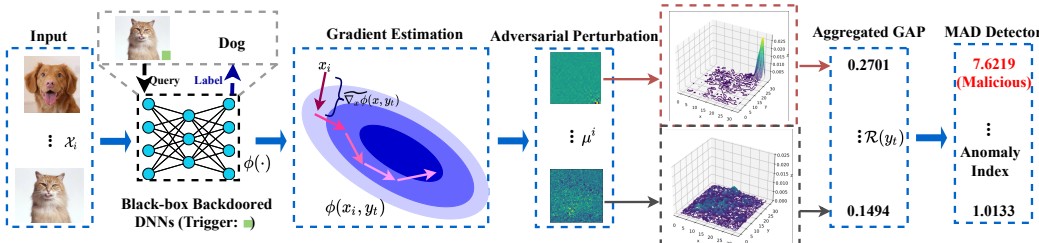

Figure 5: Overview of AEVA – Adversarial Extreme Value Analysis.

Putting all the above ideas together, we present the AEVA framework (illustrated in Fig. 5). The algorithm for computing the GAP value of a given label is described in Algorithm 1. Our approach requires a batch of legitimate samples $X_i$ for each class $i$. For each label $y_t$ to be analyzed, the algorithm will tell whether $y_t$ is backdoor-infected. We treat $y_t$ as the targeted label. Then we collect a batch of legitimate samples belonging to $y_t$, denoted as $X_t$ with the same batch size as that of $X_i$. Then, we leverage $X_t$ to make each sample within every $\{X_i\}_{i=0}^n$ to approach the decision boundary of $y_t$. This is computed by optimizing $\hat{\mu}$ from Eq. 6 using Monte Carlo gradient estimation. The adversarial peak $\mu_{max}$ is then computed for each example. The global adversarial peak (GAP) is aggregated over all labels. After calculating the GAP value $\mathcal{R}(y_i)$ for each class $i$, following previous work (Wang et al., 2019; Dong et al., 2021), we implement Median Absolute Deviation (MAD) to detect the outliers among $\{\mathcal{R}(y_i)\}_{i=1}^n$. Specifically, we use MAD to calculate the anomalous index for each $\mathcal{R}(y_i)$ by assuming $\{\mathcal{R}(y_i)\}_{i=1}^n$ fits Gaussian distribution. The outlier is then detected by thresholding the anomaly index.

## 4 EXPERIMENTS

### 4.1 SETTINGS

**Datasets.** We evaluate our approach on CIFAR-10, CIFAR-100, and Tiny-ImageNet datasets. For each task, We build 240 infected and 240 benign models with different architectures: ResNets (He et al., 2016) and DenseNets (Huang et al., 2017). We randomly select one label for each backdoor-infected model and inject sufficient poisoned samples to ensure the attack success rate $\geq 98\%$. More details can be found in the Appendix.

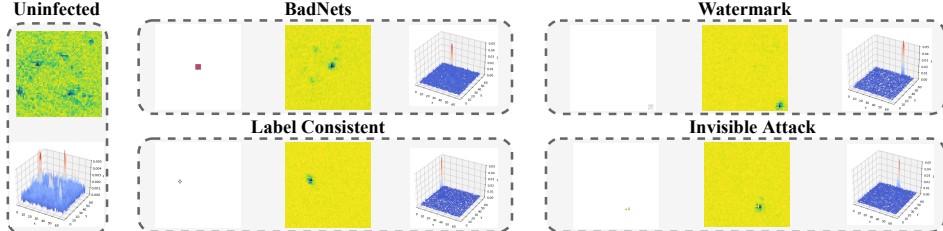

Figure 6: The visualization of adversarial perturbation $\hat{\mu}$ for infected labels with different backdoor attacks on TinyImageNet. For each attack, we show the trigger (left), $\hat{\mu}$ map (middle) and $\hat{\mu}$ distribution in 3D (right).

| Task | Detection Accuracy (ACC) | | | | | |
| --- | --- | --- | --- | --- | --- | --- |
| | BadNets | Watermark | Label Consistent | Invisible Attack | **Total(Infected)** | Benign |
| CIFAR-10 | 86.7% | 93.3% | 93.3% | 93.3% | 91.7% | 95.4% |
| CIFAR-100 | 93.3% | 100% | 96.7% | 96.7% | 96.7% | 97.5% |
| TinyImageNet | 96.7% | 100% | 93.3% | 96.7% | 96.7% | 99.2% |

Table 1: Overall performance of our approach on three tasks.

**Attack methods.** Four different attack methods are implemented: BadNets (Gu et al., 2019), Watermark attack (Chen et al., 2017), Label-Consistent Attack (Turner et al., 2019) and Invisible Attack (Li et al., 2021a). For BadNets and Watermark, the triggers are $4 \times 4$ squares. The transparency ratio for Watermark attack is 0.1. For label-consistent and invisible attacks, we use the default less-visible triggers. Around $10\%$ poison data is injected in training. Each model is trained for 200 epochs with data augmentation. More details can be found in the Appendix G.

**Baseline methods.** Since there is no existing black-box hard-label method, we compare our approach with two state-of-art white box backdoor detection methods: Neural Cleanse (NC) (Wang et al., 2019) and DL-TND (Wang et al., 2020)[2]. For each task, we use 200 samples for the gradient estimation, and the batch size for each $\{X_i\}_{i=1}^n$ is set to 40 in Algorithm 1.

**Outlier detection.** To accurately identify the anomalous $\mathcal{R}(y_i)$ among $\{\mathcal{R}(y_i)\}_{i=1}^n$, we assume the scores fit to a normal distribution and apply a constant estimator (1.4826) to normalize the computed anomaly index similar to (Wang et al., 2019). We set the threshold value $\tau = 4$ for our MAD detector, which means we identify the class whose corresponding anomaly index larger than 4 as infected. This value is chosen using a hold-out validation set of infected and benign models.

**Evaluation metrics.** We follow previous work (Kolouri et al., 2020; Wang et al., 2020; Dong et al., 2021; Guo et al., 2020) on using two common metrics: (a) *The Area under Receiver Operating Curve* (AUROC) – The Receiver Operating Curve (ROC) shows the trade-off between detection success rate for infected models and detection error rate for benign models across different decision thresholds $\tau$ for anomaly index; (b) *Detection Accuracy* (ACC) – The proportion of models are correctly identified. Regarding infected models, they are correctly identified if and only if the infected labels are identified without mistagging other uninfected labels.

## 4.2 RESULTS

We first investigate whether AEVA can reveal the singularity phenomenon of labels infected with different attack approaches. We randomly select an infected model infected by each attack approach and an uninfected model for TinyImageNet task. Notably, all these selected models are correctly identified by our approach. We plot the corresponding normalized adversarial perturbation $\mu$ in Fig. 6, which demonstrates that AEVA can accurately and distinguish the uninfected and infected labels and reveal the singularity phenomena of backdoor triggers.

Table 1 presents the overall results. AEVA can accurately detect the infected label with ACC $\geq 86.7\%$ across all three tasks and various trigger settings. We compare our approach with existing white-box detection approaches including Neural Cleanse (NC) (Wang et al., 2019) and DL-TND (Wang et al., 2020). The comparison results over all infected and benign models are shown in

---

[2]We implement NC (Wang et al., 2019) and DL-TND (Wang et al., 2020) following `https://github.com/bolunwang/backdoor` and `https://github.com/wangren09/TrojanNetDetector`

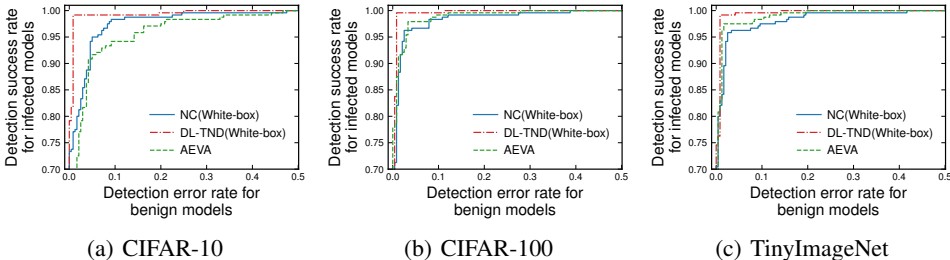

(a) CIFAR-10      (b) CIFAR-100      (c) TinyImageNet

Figure 7: The Receiver Operating Cure(ROC) for NC, DL-TND and AEVA on CIFAR-10, CIFAR-100 and TinyImageNet tasks. AEVA is black-box while the other two methods are white-box.

Fig. 7. More results can be found in Appendix J. The results suggest that AEVA achieves comparable performance with existing white-box detection approaches across different settings. Such close performance also indicates the efficacy of AEVA on black-box hard-label backdoor detection.

## 4.3 ABLATION STUDY

**The impact of trigger size.** We test AEVA on TinyImageNet with different trigger sizes for BadNets and Watermark attacks. For each trigger size and attack method, we build 60 infected models with various architectures following the configurations in Sec. 4.2. Fig. 8 shows that AEVA remains effective when trigger size is less than $14 \times 14$ with ACC $\geq 71.7\%$. For large triggers, AEVA cannot identify the infected label since the singularity property is alleviated. This is consistent with our theoretical analysis. Even though large trigger attacks can bypass AEVA, they are either visually-distinguishable or leading to a non-robust model sensitive to input change, making it less stealthy and impractical.

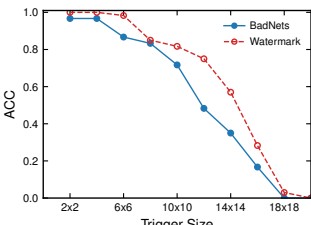

Figure 8: The impact of trigger size on detection accuracy.

**The impact of infected label numbers.** We further investigate the impact of the number of infected labels. We build 60 infected models for CIFAR-10 to evaluate AEVA. We randomly select infected labels, and we inject a $2 \times 2$ trigger for each infected label. Fig. 9 shows that AEVA performs effectively when the number of infected labels is less than 3 (*i.e.*, $30\%$ of the entire labels) with ACC $\geq 78.3\%$. It is because too many infected labels fail the MAD outlier detection. Interestingly, existing white-box detection methods (Wang et al., 2019; Chen et al., 2019b; Guo et al., 2020; Liu et al., 2019; Wang et al., 2020) cannot perform effectively in such scenarios either. However, too many infected labels would reduce the stealth of the attacker as well.

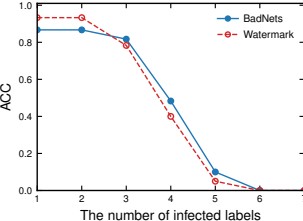

Figure 9: The impact of infected label size on detection accuracy.

Besides these studies, we further evaluate our approach under more scenarios, including the impact of the number of available labels, multiple triggers for a single infected label, different backdoor trigger shapes and potential adaptive backdoor attacks, which are included in the Appendix.

## 5 CONCLUSION

This paper takes a first step addressing the black-box hard-label backdoor detection problem. We propose the *adversarial extreme value analysis* (AEVA) algorithm, which is based on an extreme value analysis on the adversarial map, computed from the monte-carlo gradient estimation due to the black-box hard-label constraint. Extensive experiments demonstrate the efficacy of AEVA across a set of popular tasks and state-of-the-art backdoor attacks.

## ETHICS STATEMENT

Our work aims at providing neural network practitioners additional protection against backdoor attacks. We believe our work could contribute positively to the human society and avoid potential harm since it addresses a critical safety problem. We are unaware of any direct negative impact out of this work. Our method has certain limitations such as the sensitivity to large backdoor trigger and multiple infected labels. However, as we earlier in the paper, these scenarios make the attack become less stealthy and not practical in real applications.

## REPRODUCIBILITY STATEMENT

Our work is built upon comprehensive theoretical results and clear motivations. We believe the proposed method can be reproduced according to the content in the paper, *e.g.*, Algorithm 1. In addition, we have released the implementation of AEVA in `https://github.com/JunfengGo/AEVA-Blackbox-Backdoor-Detection-main`.

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

APPENDIX

# Table of Contents

## A   OTHER BACKDOOR ATTACKS

### A.1   DENSE WATERMARK BACKDOOR ATTACKS.

In short, we argue that low-magnitude dense backdoor attack(watermark) is not a practical attack method and is less concerned compared to the threat model in our paper.

First, the low-magnitude attack is a special case of our formulation where the mask m is a continuous value (here we use $a$ to represent transparency). The low-magnitude problem can be formulated as:

$\min_{a,\Delta} \sum_i \ell(\phi((1-a)x_i, a\Delta), y_t) + \lambda|a|$

This formulation does not have discrete variables so its black-box optimization is an easier task compared to our problem (where m is discrete).

Second, we will show theoretically that, in this scenario, every normal example is near the decision boundary of the network, because due to Taylor expansion and the fact that a is a small value,

$\phi((1-a)x_i + a\Delta) = \phi(x_i + a(\Delta - x_i)) \approx \phi(x_i) + a(\Delta - x_i)^\intercal \nabla_x \phi$

We know from $x_i$ to $(1-a)x_i + a\Delta$, the function output label is changed. So every example $x_i$ is at the decision boundary of the function $\phi$.

So the infected model is actually very sensitive to random noise and becomes ineffective in attacking purposes.

To empirically support this claim, we conduct experiments on CIFAR-10 using ResNet-44. The results are reported over average among ten different target labels. We select watermark triggers randomly which occupy the entire image with 0.1 transparency rate (a). We here add noise which fits uniform distribution with a low magnitude as 0.025 to each normal sample. The proportion of normal samples misclassified by the infected label is shown below. We find that such dense low-magnitude watermark triggers would make normal samples to be misclassified as the target label $y_t$ via adding small random noise. Such observations can easily identify models infected with a dense and low-magnitude watermark trigger.

- Using a global watermark trigger, $67.8\%$ of the normal samples are misclassified to other labels(Infected label)
- Using a 16x16 watermark trigger, $6.42\%$ of the normal samples are misclassified to other labels(Infected label)
- Using a 4x4 watermark trigger leads to $1.41\%$ of the normal samples misclassified
- Using a 2x2 watermark trigger leads to $0.24\%$ of the normal samples misclassified
- Training with normal examples results in $0.2\%$ of the normal samples misclassified.

### A.2   FEATURE SPACE BACKDOOR

Feature space backdoor attack which employs a generative model to transfer the style of a normal image into a trigger. However, the style transfer leads to significant visual changes which becomes less stealthy. For physical attacks like surveillance cameras, this approach is hard to implement because it requires changing the background as well. While our work is mostly focused on the input space trigger, we believe the latent space problem is beyond our threat model as well as focused problem and could be an interesting future work (for example, consider using an off-the-shelf style transfer model to evaluate the model's sensitivity towards different style effects.

## B   PROOFS

### B.1   THE PROOF OF LEMMA 1

**Lemma 1.** Given a continuous differential function $f$ defined on $\mathbb{R}^n$ and vectors $\mathbf{x}, \mathbf{h} \in \mathbb{R}^n$, there exists a real value $M \geq |\partial f/\partial h_i| \forall 1 \leq i \leq n$ such that

$$|f(\mathbf{x} + \mathbf{h}) - f(\mathbf{x})| \leq M\|\mathbf{h}\|_1 \,. \tag{8}$$

**Proof of Lemma 1:**   The first order Taylor expansion gives

$$f(\mathbf{x} + \mathbf{h}) = f(\mathbf{x}) + R(\mathbf{h}) \tag{9}$$

where $R(\mathbf{h})$ is the Lagrange remainder, *i.e.*,

$$R(\mathbf{h}) = \sum_{i=1}^{n} \frac{\partial f(\mathbf{x} + c\mathbf{h})}{\partial x_i} \cdot h_i, \quad \text{for some } c \in (0, 1). \tag{10}$$

Let $M \geq |\partial f(\mathbf{x})/\partial x_i| \, \forall \mathbf{x}$, we have

$$|R(\mathbf{h})| \leq M \left| \sum_{1}^{n} h_i \right| = M\|\mathbf{h}\|_1 \, . \tag{11}$$

So, we have

$$|f(\mathbf{x} + \mathbf{h}) - f(\mathbf{x})| \leq M\|\mathbf{h}\|_1 \, . \tag{12}$$

$\square$

## B.2   THE PROOF OF LEMMA 2 – A THEORY ON THE LINEAR CASE

We perform a theoretical study of linear models to shed light about our empirical finding. Here, we restate Lemma 3 below for convenience.

**Lemma 2.**   Given a linear model parameterized by $\theta$ optimized on $N_o$ original training examples and $N_b$ backdoored training examples with an objective in Eq. 2 and a mean-squared-error loss, the adversarial solution $\hat{\mu}$ optimized using Eq. 6 will be dominated by input dimensions corresponding to backdoor mask $m$ when $N_b$ is large, *i.e.*,

$$\lim_{N_b \to \infty} \frac{\|(1 - m) \odot \hat{\mu}\|_1}{\|\hat{\mu}\|_1} = 0 \, . \tag{13}$$

**Proof of Lemma 2:**   To ease the reading of notations, we use $(\cdot)_{a \times b}$ to represent matrix with $a$ rows and $b$ columns, respectively. Assuming $\phi(x; \theta) = \theta^{\mathsf{T}} x$ is a linear model whose output is a floating number, and the loss function is $\ell^2$, i.e., $\ell(x, y) = (x - y)^2$. Then Eq. 2 becomes a linear regression problem and can be represented in the matrix form of least square, *i.e.*,

$$F = \|X\theta - Y\|^2 + \|\hat{X}\theta - \hat{Y}\|^2 \tag{14}$$

where $\hat{x}_i = (1 - m) \odot x'_i + m \odot \Delta$ is an backdoor example converted from a normal example $x'_i$ and $\hat{y}_i = y_t$. $X$ is the matrix composed of row vectors $x_i$ and $Y$ is the matrix composed of row vectors $y_i$. The solution of this least square can be found by setting gradient to zero, such that

$$\hat{\theta} = (X^{\mathsf{T}} X + \hat{X}^{\mathsf{T}} \hat{X})^{-1} (X^{\mathsf{T}} Y + \hat{X}^{\mathsf{T}} \hat{Y}) \, . \tag{15}$$

It is known that a normal training without backdoors will lead to the pseudo-inverse solution $\theta = (X^{\mathsf{T}} X)^{-1} X^{\mathsf{T}} Y$. Without loss of generality, we assume the backdoor trigger occupies the last $q$ dimensions. And the first $p$ dimensions are normal values, *i.e.*, $q = d - p + 1$ where $d$ is the total number of dimensions. So, we can rewrite the input $X = [X_{1:p}, X_{p:d}]$ and the mask $m = [\mathbf{0}_{1 \times p}, \mathbf{1}_{1 \times (d-p+1)}]$. Let $\boldsymbol{\delta}$ be the sub-vector of $\Delta$ such that $\boldsymbol{\delta} = \Delta_{p+1:d}$, a $q$ dimensional row vector. Then we have $\hat{X} = [X_{1:p}, \mathbf{1}_{N_b \times 1} \boldsymbol{\delta}]$ where $N_b$ is the total number of backdoor examples. Then,

$$\hat{X}^{\mathsf{T}} \hat{X} = \begin{bmatrix} X_{1:p}^{\mathsf{T}} X_{1:p} & 0 \\ 0 & N_b \boldsymbol{\delta}^{\mathsf{T}} \boldsymbol{\delta} \end{bmatrix} \, . \tag{16}$$

Similarly, we find $\hat{Y} = \mathbf{1} y_t$ and assume zero mean of the input, i.e., $\sum x_i = 0$, then we have

$$\hat{X}^{\mathsf{T}} \hat{Y} = \begin{bmatrix} \mathbf{0}_{p \times 1} \\ N_b y_t \boldsymbol{\delta}^{\mathsf{T}} \end{bmatrix} \, . \tag{17}$$

While $N_b$ becomes larger, both $\hat{X}^\mathsf{T}\hat{X}$ and $\hat{X}^\mathsf{T}\hat{Y}$ increase significantly. When $N_b \to \infty$, Eq. 15 becomes

$$\hat{\theta} \to (\hat{X}^\mathsf{T}\hat{X})^{-1}(\hat{X}^\mathsf{T}\hat{Y}) = \begin{bmatrix} \mathbf{0}_{p\times 1} \\ y_t(\boldsymbol{\delta}^\mathsf{T}\boldsymbol{\delta})^{-1}\boldsymbol{\delta}^\mathsf{T} \end{bmatrix} . \tag{18}$$

In another word, we know the normalized values in the non-masked area of $\theta$ approach 0 such that

$$\lim_{N_b \to \infty} \frac{\|(1-m)\odot\hat{\theta}\|_1}{\|\hat{\theta}\|_1} = 0 . \tag{19}$$

Now, let us look at the adversarial objective in Eq. 8, which is equivalent to

$$G = \|\theta^\mathsf{T}(x+\mu) - y_t\|^2 , \tag{20}$$

the gradient of which is

$$\nabla_\mu G = 2(\theta^\mathsf{T}(x+\mu) - y_t)\theta . \tag{21}$$

Since $2(\theta^\mathsf{T}(x+\mu)-y_t)$ is a scalar and each of the gradient updates is conducted by $\mu = \mu + \lambda\nabla_\mu G$, we know no matter how many gradient update steps are applied, $\mu$, initialized at 0, always moves in the exact same direction of $\theta$, $i.e.$, $\hat{\mu} \parallel \theta$. In this case, $\hat{\mu}$ will also be dominant by the last $q$ dimensions, or the dimensions corresponded to the mask $m$. In another word, we have

$$\lim_{N_b \to \infty} \frac{\|(1-m)\odot\hat{\mu}\|_1}{\|\hat{\mu}\|_1} = 0 . \tag{22}$$

$\square$

### B.3   PROOF OF LEMMA 3 – AN ANALYSIS FOR KERNEL REGRESSION

**Lemma 3.** Suppose the training dataset consists of $N_o$ original examples and $N_b$ backdoor examples, i.i.d. sampled from uniform distribution and belonging to $k$ classes. Each class contains equally $N_o/k$ normal examples. Let $\phi$ be a multivariate kernel regression with the objective in Eq. 2, an RBF kernel $K(\cdot,\cdot)$ Then the adversarial solution $\hat{\mu}$ to Eq. 6 under cross-entropy loss $\ell(\hat{y},y) = -\sum_i y_i \log \hat{y}_i$ ($y \in \{0,1\}^k$ is the one-hot label vector) should satisfy that

$$\lim_{N_b \to \infty} \mathbb{E}[(1-m)\odot\hat{\mu}] = \mathbf{0} . \tag{23}$$

**Proof of Lemma 3:** The output of $\phi$ is a $k$ dimensional vector. Let us assume $\phi_t(\cdot) \in \mathbb{R}$ be the output corresponding to the target class $t$. We know the kernel regression solution is

$$\phi_t(\cdot) = \frac{\sum_{i=1}^{N_o} K(\cdot, x_i)y_{i,t} + \sum_{i=1}^{N_b} K(\cdot, \hat{x}_i)\hat{y}_{i,t}}{\sum_{i=1}^{N_o} K(\cdot, x_i) + \sum_{i=1}^{N_b} K(\cdot, \hat{x}_i)} \tag{24}$$

where $\hat{x}_i = (1-m)\odot x_i + m\odot\Delta$ is the backdoored example and $\hat{y}_i$ is the corresponding one-hot label (so we always have $\hat{y}_{i,t} = 1$). Because the examples are evenly distributed, there are $N/k$ examples labeled positive for class $t$. Without loss of generality, we assume $y_{j,t} = 1$ when $j \in [1, N_o/k]$ and $y_{j,t} = 0$ otherwise. Then the regression solution becomes

$$\phi_t(\cdot) = \frac{\sum_{i=1}^{N_o/k} K(\cdot, x_i) + \sum_{i=1}^{N_b} K(\cdot, \hat{x}_i)}{\sum_{i=1}^{N_o} K(\cdot, x_i) + \sum_{i=1}^{N_b} K(\cdot, \hat{x}_i)} \tag{25}$$

Please note the regression above is derived with a mean squared loss. In the adversarial analysis Eq. 6, we can use any alternative loss function. In this Lemma, we assume the loss in the adversarial analysis is cross-entropy, which is a common choice. So, we have

$$\ell(\phi(x), y_t) = -\log\phi_t(x) \tag{26}$$

$$= -\log(S_t + \hat{S}) + \log(S + \hat{S}) \tag{27}$$

where $S_i(\cdot) = \sum_{y_j=i} K(\cdot, x_j)$ and $S = S_1 + S_2 + \ldots + S_k$ and $\hat{S} = \sum_{i=1}^{n_b} K(\cdot, \hat{x}_i)$.

The derivative of loss w.r.t. $x$ becomes

$$\frac{\partial \ell(\phi(x))}{\partial x} = -\frac{1}{S_t + \hat{S}} \frac{\partial (S_t + \hat{S})}{\partial x} + \frac{1}{S + \hat{S}} \frac{\partial (S + \hat{S})}{\partial x} \tag{28}$$

By using gradient descent, $\hat{\mu}$ moves along the negative direction of the loss gradient, *i.e.*,

$$\Delta \mu = \frac{1}{S_t + \hat{S}} \frac{\partial (S_t + \hat{S})}{\partial x} - \frac{1}{S + \hat{S}} \frac{\partial (S + \hat{S})}{\partial x} \tag{29}$$

$$= \frac{\partial \hat{S}}{\partial x} \left( \frac{1}{S_t + \hat{S}} - \frac{1}{S + \hat{S}} \right) + \frac{\partial S_t}{\partial x} \left( \frac{1}{S_t + \hat{S}} - \frac{1}{S + \hat{S}} \right) - \frac{1}{S + \hat{S}} \frac{\partial (S - S_t)}{\partial x} \tag{30}$$

$$= a \frac{\partial \hat{S}}{\partial x} + a \frac{\partial S_t}{\partial x} - b \frac{\partial (S - S_t)}{\partial x} \tag{31}$$

where $a, b$ are both positive scalar values. When $N_b$ becomes large, the first term dominates the gradient direction, *i.e.*, $\Delta \mu \propto \frac{\partial \hat{S}}{\partial x}$.

Since we assume the kernel $K(x, x') = \exp(-\gamma \|x - x'\|^2)$ is RBF, then we have its derivative

$$\frac{\partial K(x, x')}{\partial x} = -2\gamma K(x, x')(x - x') \tag{32}$$

Then we have

$$\frac{\partial \hat{S}}{\partial x} = \sum_{i=1}^{N_b} \frac{\partial K(x, \hat{x}_i)}{\partial x} \tag{33}$$

$$= \sum_{i=1}^{N_b} -2\gamma K(x, \hat{x}_i)(x - \hat{x}_i) \tag{34}$$

Without loss of generality, we assume the mask $m = [\mathbf{0}_{1 \times p}, \mathbf{1}_{1 \times (d-p+1)}]$ (where first $p$ dimensions are 0s and the remaining are 1s). Then the first $p$ dimensions of $\hat{x}_i$ are the same as those of $x_i$, while the remaining dimensions equivalent to those of $\Delta$.

**Case 1:** $p + 1 \leq j \leq d$. The $j$-th dimension of $\frac{\partial \hat{S}}{\partial x}$ can be written as

$$\left( \frac{\partial \hat{S}}{\partial \mu} \right)_j = \left( -2\gamma \sum_{i=1}^{N_b} K(z + \mu, \hat{x}_i) \right) (z_j + \mu_j - \delta_j) \tag{35}$$

So at convergence, $\mu_j = \delta_j - z_j$ for the last $d - p + 1$ dimensions. The expectation is $\mathbb{E}[\mu_j] = \delta_j - \mathbb{E}[z_j]$.

**Case 2:** $1 \leq j \leq p$. For the first $p$ dimensions, we have

$$\left( \frac{\partial \hat{S}}{\partial \mu} \right)_j = -2\gamma \sum_{i=1}^{N_b} K(z + \mu, \hat{x}_i)(z_j + \mu_j - x_{ij}) \tag{36}$$

$$= -2\gamma \sum_{i=1}^{N_b} e^{-\gamma \|z + \mu - \hat{x}_i\|^2} (z_j + \mu_j - x_{ij}) . \tag{37}$$

Since $\mu_s + z_s = \delta_s = \hat{x}_{is} = x_{is}$ for $p + 1 \leq s \leq d$, then we have

$$\left( \frac{\partial \hat{S}}{\partial \mu} \right)_j = -2\gamma \sum_{i=1}^{N_b} e^{-\gamma \sum_{1 \leq v \leq p} (z_v + \mu_v - x_{iv})^2} (z_j + \mu_j - x_{ij}) \tag{38}$$

Since $N_b$ is large and $\hat{x}_i$ is i.i.d., then the summation can be approximated with integration, *i.e.*,

$$\left( \frac{\partial \hat{S}}{\partial \mu} \right)_j = -2\gamma \int_x e^{-\gamma \sum_{1 \leq v \leq p} (z_v + \mu_v - x_{iv})^2} (z_j + \mu_j - x_j) dx . \tag{39}$$

When $\gamma$ is large, the integration is the expectation of the offset to the center of a Gaussian distribution (centered at $z_j + \mu_j$). Due to symmetry, the derivative becomes 0 for most of the $\mu$ locations. This would be the common cases because a smaller $\gamma$ means examples with longer distance can more significantly influence the function output (which is not desired in a classifier with high accuracy).

Nonetheless, in rare cases when the $\gamma$ becomes small, we can still find the stationary point of the derivative. Given the fact that $e^{-\gamma \sum_{1 \le v \le p} (z_v + \mu_v - x_{iv})^2}$ is always positive, we know $\left( \frac{\partial \hat{S}}{\partial \mu} \right)_j = 0 \implies z_j + \mu_j - x_j = 0$. But it is impossible to reach. Again, utilizing symmetry of the normal distribution and the fact that $x$ is i.i.d. uniform, we find when $z_j + \mu_j - \mathbb{E}[x_j] < 0$, the derivative is positive; and when $z_j + \mu_j - \mathbb{E}[x_j] > 0$, the derivative is negative. Since we use gradient descent to find the optimal $\hat{\mu}$. We can conclude that, at the convergence, $z_j + \mu_j = \mathbb{E}[x_j]$. Then $\mathbb{E}[\mu_j] = \mathbb{E}[x_j] - \mathbb{E}[z_j] = 0$ because $z$ is sampled from the same distribution of $x$.

To summarize,

$$\mathbb{E}[\hat{\mu}_j] = \begin{cases} 0, & 1 \le j \le p \\ \delta_j - \mathbb{E}[z_j], & p < j \le d \end{cases} \tag{40}$$

Then we know, as $N_b \to \infty$,

$$\mathbb{E}[(1 - m) \odot \mu] \to 0 \tag{41}$$

$\square$

## C  MONTE CARLOS GRADIENT ESTIMATION

Leveraging the Monte Carlo based estimation to craft target adversarial perturbations for the targeted label $y_t$ typically requires $x$ and $x_t$, where $x_t$ is a legitimate sample satisfies $f(x_t) = y_t$. Firstly, $x$ is forced to approach the decision boundary of $f(\cdot; \theta)$ for the target label $y_t(\phi(x, y_t) \to 0.5)$, which stimulates the efficiency and accuracy of gradient estimation. We can make $\phi(x, y_t) \to 0.5$ through projection process:

$$x \leftarrow (1 - \alpha)x + \alpha x_t, \tag{42}$$

$\alpha \in (0, 1)$ is a parameters for projection. During each gradient estimation procedure, $\alpha$ is set through binary search (Chen et al., 2020a).

After that, we leverage Monte Carlo sampling to estimate $\widetilde{\nabla \phi}_x(x, y_t)$, its procedure can be expressed as:

$$\widetilde{\nabla_x \phi}(x, y_t) = \frac{1}{N} \sum_{i=0}^{N} \mathcal{S}(x + \delta \mu'_i, y_t) \mu'_i, \tag{43}$$

where $\{\mu'\}_{i=0}^{N}$ are perturbations *i.i.d* sampled from uniform distribution and $\delta$ is a small positive value representing the magnitude of perturbation. $\mathcal{S}$ is an indicator function such that

$$\mathcal{S}(x, y) = \begin{cases} 1, & f(x) = y, \\ -1, & f(x) \neq y. \end{cases} \tag{44}$$

To further eliminate the variance induced by Monte Carlo sampling, we can improve $\widetilde{\nabla \phi}_x(x, y_t)$ via:

$$\widetilde{\nabla \phi}_x(x, y_t) = \frac{1}{N} \sum_{i=0}^{N} \{ \mathcal{S}(x + \delta \mu'_i, y_t) - \frac{1}{N} \sum_{i=0}^{N} \mathcal{S}(x + \delta \mu'_i, y_t) \} \mu'_i \tag{45}$$

Using *cosine* angle to measure the similarity between $\widetilde{\nabla_x \phi}(x, y_t)$ and $\nabla_x \phi(x, y_t)$, previous work (Chen et al., 2020a) have proved that:

$$\lim_{\delta \to 0} cos\angle(\mathbb{E}[\widetilde{\nabla_x \phi}(x, y_t)], \nabla_x \phi(x, y_t)) = 1 , \tag{46}$$

thus a smaller $\delta$ (0.01) is selected for ensuring the efficacy of gradient estimation.

Regarding the constraints on $||\mu||_1$, we further processed the estimated gradient via:

$$\widetilde{\nabla_x \phi}(x, y_t) \leftarrow \frac{\widetilde{\nabla_x \phi}(x, y_t)}{||\widetilde{\nabla_x \phi}(x, y_t)||_1} \ . \tag{47}$$

We refer readers to (Chen et al., 2020a) for the complete algorithm for gradient estimation.

## D  ALGORITHM FOR COMPUTING AGGREGATED GLOBAL ADVERSARIAL PEAK (GAP)

---
**Algorithm 1** Aggregated Global Adversarial Peak (GAP)

---
1: **Input:** Targeted DNN $f(\cdot; \theta)$; label to be analyzed $y_t$; legitimate input batches $\{X_i\}_{i=1}^n$ evenly sampled across classes; the targeted input batch $X_t$;
2: **Output:** Aggregated GAP value $\mathcal{R}(y_t)$;

3: Initialize $\mathcal{R}(y_t) = 0$
4: **for** $i = 1, \ldots, t-1, t+1, \ldots, n$ **do**
5:     **for** $j = 1, \ldots, \text{len}(X_i)$ **do**
6:         Solve $\mu_j^i = \arg\min_\mu \ell(\phi(x_{ij} + \mu, y_t; \theta), y_t) + \lambda||\mu||_1$ using MC gradient estimation;
7:     **end for**
8:     Find the GAP value for the current label: $\mu_{max}^i = \max_{j,u,v} \mu_{j,u,v}^i$;
9:     Aggregate GAP values over all labels: $\mathcal{R}(y_t) = \mu_{max}^i + \mathcal{R}(y_t)$;
10: **end for**
11: **Return:** $\mathcal{R}(y_t)$

---

## E  THE DETAILED CONFIGURATIONS FOR EXPERIMENTAL DATASETS AND TRIGGERS

| Task | # labels | Input size | # training images |
|------|----------|------------|-------------------|
| CIFAR-10 | 10 | 32x32 | 50000 |
| CIFAR-100 | 100 | 32x32 | 50000 |
| TinyImageNet | 200 | 64x64 | 1000000 |

Table 2: Detailed information about dataset for each task.

The detailed information for each task is included in the Table. E. The data augmentation utilized for building each model is following `https://keras.io/zh/examples/cifar10_resnet/`.

## F  THE ACCURACY AND ATTACK SUCCESS RATE(ASR) FOR EVALUATED MODELS

The accuracy and ASR for the evaluated models for each task in included in Table. 3.

| Task | Infected Model | | Normal Model Accuracy |
|------|----------|------|-----------------------|
| | Accuracy | ASR | |
| CIFAR-10 | $\geq 90.04\%$ | $\geq 97.7\%$ | $\geq 92.31\%$ |
| CIFAR-100 | $\geq 69.17\%$ | $\geq 96.27\%$ | $\geq 71.41\%$ |
| TinyImageNet | $\geq 58.98\%$ | $\geq 97.22\%$ | $\geq 60.11\%$ |

Table 3: Accuracy and ASR for the evaluated models for each task

## G  THE AMOUNTS OF EVALUATED MODELS FOR VARIOUS TASK AND ATTACK APPROACHES

For each attack and task, we build 60 infected models and 60 uninfected models, respectively. Notably, the uninfected models for different attack approaches are randomly selected and different. Each uninfected and infected model sets are built evenly upon ResNet-18, ResNet-44, ResNet-56, DenseNet-33, DenseNet-58 these five models.

## H  DETAILS OF VISUALIZATION PROCESS

---

**Algorithm 2** Visualize $\mu$

---

1: **Input:** Targeted DNN $f(\cdot;\theta)$; label to be analyzed $y_t$; legitimate input batches $\{X_i\}_{i=1}^n$ evenly sampled across classes; the targeted input batch $X_t$;
2: **Initialize** $\mu$
3: **Output:** Normalized $\mu$;
4: **for** $i = 1, \ldots, t-1, t+1, \ldots, n$ **do**
5:     **for** $j = 1, \ldots, \text{len}(X_i)$ **do**
6:         Solve $\mu_j^i = \arg\min_\mu \ell(\phi(x_{ij} + \mu, y_t; \theta), y_t) + \lambda\|\mu\|_1$ using MC gradient estimation;
7:     **end for**
8:     Find $\mu^i$ owns the GAP value for the current label: $\mu^i = \arg\max_{\mu_j^i} \mu_{j,u,v}^i$;
9:     Aggregate $\mu^i$ over all labels: $\mu = \mu^i + \mu$;
10: **end for**
11: **Return:** $\mu/\|\mu\|_1$

---

## I  DETAILED ROC FOR VAROUS TASKS AND ATTACK APPROACHES

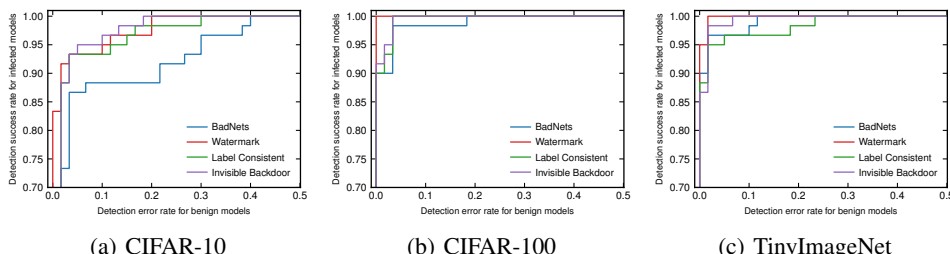

(a) CIFAR-10      (b) CIFAR-100      (c) TinyImageNet

Figure 10: The Receiver Operating Cure(ROC) for AEVA on CIFAR-10, CIFAR-100 and TinyImageNet tasks.

## J  THE COMPARISON RESULTS BETWEEN AEVA AND NC, DL-TND

| Attack | Method | Detection Results | |
|---|---|---|---|
| | | AUROC | ACC |
| BadNets | NC (Wang et al., 2019) | 0.976 | 95.0% |
| | DL-TND (Wang et al., 2020) | 0.999 | 99.2% |
| | AEVA (Ours) | 0.930 | 91.5% |
| Watermark | NC (Wang et al., 2019) | 0.983 | 95.8% |
| | DL-TND (Wang et al., 2020) | 1 | 100% |
| | AEVA (Ours) | 0.968 | 94.4% |
| Label Consistent | NC (Wang et al., 2019) | 0.961 | 94.2% |
| | DL-TND (Wang et al., 2020) | 0.992 | 98.3% |
| | AEVA (Ours) | 0.968 | 94.4% |
| Invisible Attack | NC (Wang et al., 2019) | 0.980 | 95.8% |
| | DL-TND (Wang et al., 2020) | 0.999 | 99.1% |
| | AEVA (Ours) | 0.970 | 94.4% |

Table 4: The two metrics for backdoor detection on the CIFAR-10 task using three backdoor detection methods: NC, DL-TND, and AEVA. Higher values in AUROC and ACC are better.

| Attack | Method | Detection Results | |
| --- | --- | --- | --- |
| | | AUROC | ACC |
| BadNets | NC (Wang et al., 2019) | 0.978 | 95.8% |
| | DL-TND (Wang et al., 2020) | 0.999 | 99.1% |
| | AEVA (Ours) | 0.973 | 95.4% |
| Watermark | NC (Wang et al., 2019) | 0.961 | 94.2% |
| | DL-TND (Wang et al., 2020) | 1 | 100% |
| | AEVA (Ours) | 0.992 | 98.8% |
| Label Consistent | NC (Wang et al., 2019) | 0.964 | 94.2% |
| | DL-TND (Wang et al., 2020) | 0.999 | 99.1% |
| | AEVA (Ours) | 0.981 | 96.8% |
| Invisible Attack | NC (Wang et al., 2019) | 0.989 | 97.5% |
| | DL-TND (Wang et al., 2020) | 1 | 100% |
| | AEVA (Ours) | 0.984 | 97.1% |

Table 5: The two metrics for backdoor detection on the CIFAR-100 task using three backdoor detection methods: NC, DL-TND, and AEVA. Higher values in AUROC and ACC are better.

| Attack | Method | Detection Results | |
| --- | --- | --- | --- |
| | | AUROC | ACC |
| BadNets | NC (Wang et al., 2019) | 0.956 | 91.6% |
| | DL-TND (Wang et al., 2020) | 0.999 | 99.1% |
| | AEVA (Ours) | 0.988 | 97.9% |
| Watermark | NC (Wang et al., 2019) | 0.972 | 94.2% |
| | DL-TND (Wang et al., 2020) | 1 | 100% |
| | AEVA (Ours) | 0.999 | 99.6% |
| Label Consistent | NC (Wang et al., 2019) | 0.983 | 96.7% |
| | DL-TND (Wang et al., 2020) | 0.999 | 99.1% |
| | AEVA (Ours) | 0.983 | 96.7% |
| Invisible Attack | NC (Wang et al., 2019) | 0.982 | 96.7% |
| | DL-TND (Wang et al., 2020) | 0.999 | 99.1% |
| | AEVA (Ours) | 0.986 | 97.9% |

Table 6: The two metrics for backdoor detection on the Tiny Imagenet task using three backdoor detection methods: NC, DL-TND, and AEVA. Higher values in AUROC and ACC are better. Each approach are evaluated using 60 infected and 60 benign models.

## K EXPERIMENTS FOR DIFFERENT TRIGGERS

### K.1 EVALUATION ON DYNAMIC AND SPARSE BUT NOT COMPACT TRIGGERS

We evaluate AEVA for CIFAR-10 task under the dynamic and non-compact triggers scenario. We randomly generate three non-compact triggers(shown in Figure. 11) and perform backdoor attacks against a randomly selected label using these dynamic triggers. We test AEVA using 10 different DNN models, which are evenly built upon ResNet-18, ResNet-44, ResNet-56, DenseNet-33 and DenseNet-58.

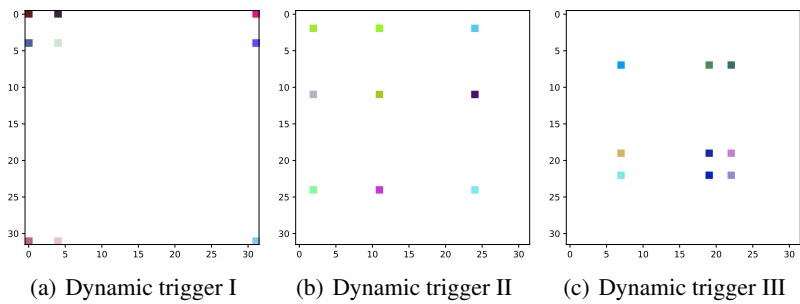

(a) Dynamic trigger I     (b) Dynamic trigger II     (c) Dynamic trigger III

Figure 11: Dynamic Triggers

AEVA successfully identify all models embedded with dynamic backdoors and the corresponding infected labels. The detailed Aggregated Global Adversarial Peak(AGAP) values and Anomaly Index given by the MAD outlier detection for infected and uninfected labels are shown in Fig. 12. We find AEVA predicts all backdoored models as infected with Anomaly Index $\geq 4$. Such results demonstrate that AEVA is resilient to dynamic and sparse but not compact triggers.

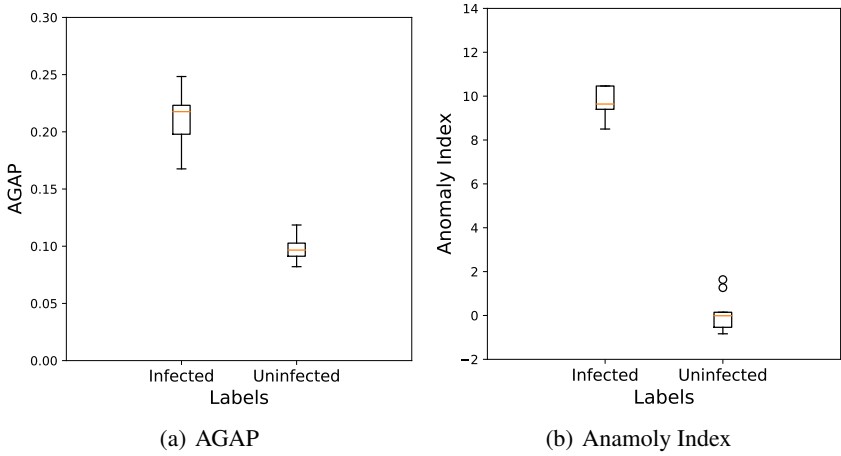

(a) AGAP

(b) Anamoly Index

Figure 12: AGAP and Anomaly Index for infected and uninfected labels

We also evaluate AEVA on three different triggers with different shapes, which are shown in Fig. 13. For each trigger, we build 30 models on TinyImageNet. Each set of models are evenly built upon ResNet-18, ResNet-44, ResNet-56, DenseNet-33, DenseNet-58 these architectures. The results are shown in Table. 7.

| Attack Approach | ACC |
|---|---|
| Trigger I | 93.3% |
| Trigger II | 93.3% |
| Trigger III | 90.0% |

Table 7: Results for other different triggers

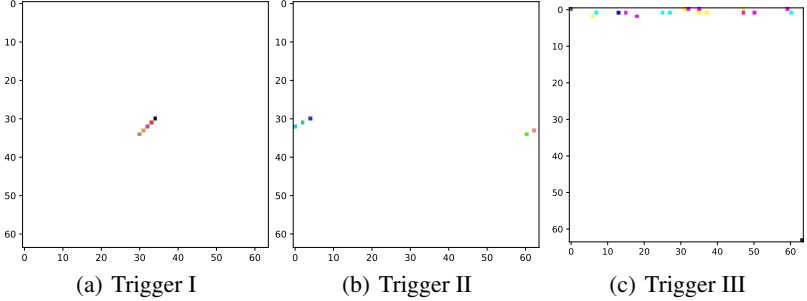

(a) Trigger I

(b) Trigger II

(c) Trigger III

Figure 13: Triggers with different shapes

## L    RESULTS FOR IMPACT OF THE NUMBER OF IMAGES PER LABELS.

We further explore how many samples for each label are required by AEVA to be effective. We test AEVA on TinyImageNet and the experimental configurations are consistent with Sec.4.1(*i.e.*, 240

infected and 240 benign models). Our results are illustrated in the Fig. 14, which plots the metrics against the number of samples per class. We find that our approach requires more samples (32) for each class to achieve optimal performance.

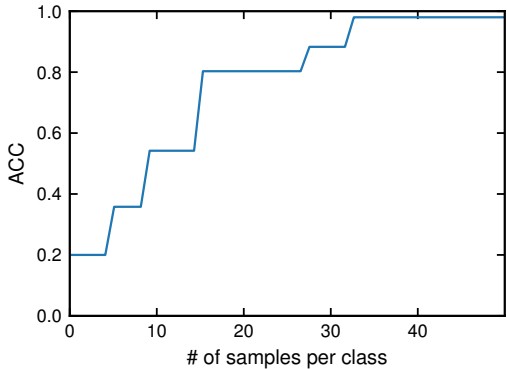

Figure 14: The impact for the number of samples for each class

## M    THE IMPACT OF THE NUMBER OF THE LABELS FOR AVAILABLE IMAGES.

Since our approach aims to detect backdoors in the black-box setting, a realistic scenario is that while determining if a specific label is infected, the clean data from all other labels may be unavailable to the defender. So, in this section, we investigate the impact of the number of labels for which clean data is available for our approach. We test our approach on TinyImageNet using 60 models infected with BadNets ($4 \times 4$ squares) following the configurations of Sec. 4.2. We vary the number of uninfected labels for available samples between [1, 200], randomly sampled from the 200 labels available in the TingImageNet dataset. Figure. 15 illustrates our results by plotting the metrics against the number of available labels. We observe that even with eight available labels, our approach can still detect backdoors in most cases, i.e., with merely $4\%$ of the labels being available, our approach is still viable. This study exhibits our approach's practicality, which can work in the black-box setting and on relatively small available data. Notably, with fewer labels AEVA will always correctly tag the uninfected models. This is because few labels will cause the anomaly index low thus resilient to uninfected models.

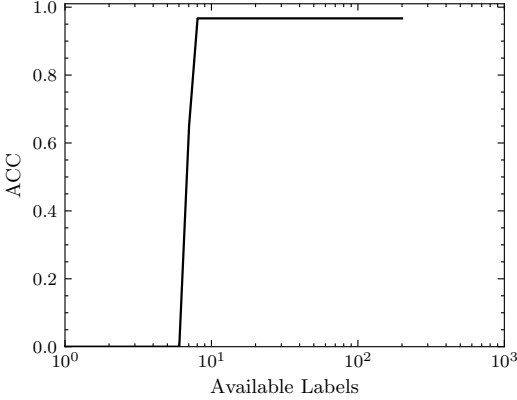

Figure 15: The impact for the number of samples for each class

## N    MULTIPLE TRIGGERS WITHIN THE SINGLE INFECTED LABEL SCENARIOS

We also investigate the impact caused by multiple triggers within the single infected label. We here choose different $4 \times 4$ squares located at different places as the triggers implemented following

BadNets. We randomly select a label as the infected label. We built 60 infected models with different architectures(*i.e.*, ResNet-18, ResNet-44, ResNet-56, DenseNet-33, DenseNet-58). The results for TinyImageNet are shown in Table. 8. Notably, for TinyImageNet, injecting too many triggers ($\geq 4$) in the single label would cause the target model's accuracy drop (i.e., $\geq 3.7\%$), which is inconsistent with the threat model for backdoor attacks (Gu et al., 2019; Chen et al., 2017; Liu et al., 2018).

As seen in Table. 8, our approach can still perform effective when injecting two $4 \times 4$ triggers in the single infected label. However, when the number of injected triggers becomes more than 2, our approach becomes less effective. This should be caused by that multiple-triggers would reduce the singularity properties for the adversarial perturbations. To address this issue, we select the sum of the largest five points as the $\mu_{max}$ instead. The results are shown in Table. 9. By selecting more points to calculate the $\mu_{max}$, AEVA can still perform effective under the multiple triggers within the single label scenarios and have no impact on the detection accuracy for uninfected models.

| Attack Approach | ACC |
|---|---|
| Two infected triggers | 81.7% |
| Three infected triggers | 48.3% |

Table 8: Results for multiple triggers within the single infected label

| Attack Approach | ACC |
|---|---|
| Two infected triggers | 93.3% |
| Three infected triggers | 88.3% |

Table 9: Results II for multiple triggers within the single infected label

## O    POTENTIAL ADAPTIVE BACKDOOR ATTACKS

We here consider two potential backdoor attacks which can bypass AEVA.

### O.1    ATTACKS WITH MULTIPLE TARGET LABELS

Since AEVA is sensitive to the number of infected labels, the attacker can infect multiple labels with different backdoor triggers. Indeed, making multiple labels (*i.e.*$\geq 30\%$;) infected will make the anomaly indexes produced by the MAD detector significantly drop. However, such attack can only be successfully implemented for some small datasets which own a few labels(e.g., CIFAR-10, etc). Reported by (Wang et al., 2019), the state-of-the-art model (DeepID) for Youtube Face dataset can not maintain the average attack success rate and model accuracy at the same time when more than $15.6\%$ labels are infected. In another word, when multiple labels are infected, the infected model's accuracy is likely to get worse if the attacker wants to keep attack success rate. So too many infected labels will reduce the stealth of the attack. We conducted experiments on TinyImageNet, which reveals that when over 14 labels are infected with different triggers, the model accuracy (ResNet-44 and DenseNet-58) decreases to $\leq 57.61\%$ (around 3% lower than a normal model) when preserving the attack effectiveness (ASR $\geq 97.12\%$) for each infected label.

### O.2    ATTACK WITH LARGE TRIGGERS

Another potential adaptive backdoor attack is that the attacker would implement a dense backdoor trigger which would alleviate the singularity phenomenon. As we claim in Section 4.3, a dense backdoor trigger would appear visually-distinguishable to the human beings which perform less stealthy, that is also reported by Neural Cleanse (Wang et al., 2019). Indeed, there exists an interesting exceptional watermark attacks. However, we empirically prove that such dense watermark attack (Chen et al., 2017) would make infected DNNs non-robust which would make the inputs sensitive to the random noise. The details are included in Appendix A.

