# OpenReview forum: "AEVA: Black-box Backdoor Detection Using Adversarial Extreme Value Analysis"
_ICLR.cc/2022/Conference — ICLR 2022 Poster_

### Official Review · Reviewer_hfE9 · 2021-10-28

**Correctness:** 3
**Technical Novelty And Significance:** 3
**Empirical Novelty And Significance:** 3
**Recommendation:** 8
**Confidence:** 4

**Main Review:**


Pros
1.	This paper is well-written and easy to follow.
2.	The topic is of great significance and sufficient interest to ICLR audiences. In particular, black-box hard-label backdoor detection is probably the hardest problem in backdoor defense and has the highest commercial potential. As such, I think this paper should be encouraged, although it still has some problems.
3.	The authors provide some theoretical supports for their method and open-source their codes, which should be encouraged.
4.	I appreciate that the adaptive attacks and potential limitations are also included in the paper. It can prevent readers from being overly optimistic about safety.


In general, I enjoy the reading of this paper and I think the proposed method is also moderately novel. However, I still have some concerns. I will increase my score if the authors can address my concerns. The detailed comments are as follows:


Major Comments
1.	My main concern lies in the novelty of Lemma 2-3. The optimization problem (6) seems to be a classical problem. As such, I have to worry about whether its analysis (i.e., Lemma 2-3) is not new. I will not decrease my score even Lemma 2-3 are not new if the author can provide proper references and illustrations.
2.	I would like to see the results of defending against attacks with dynamic and sparse but not compact triggers.
3.	Please analyze the effects of key hyper-parameters (e.g., lambada) involved in the proposed method.
4.	Please provide more details and results in the analysis of potential adaptive attacks.


Minor Comments
1.	Please double check the reference format, especially that of conference papers. Please cite the official version of all papers (e.g., BadNets should be IEEE ACCESS, DL-TND should be ECCV).


Note: I didn't check all proofs carefully. But the lemma seems reasonable and consistent with the experimental results.


Post-rebuttal Comments:
I would like to thank the authors for their clarifications. Since most of my concerns have been addressed, I increase my score to 8.
PS: It would be better if you can point out whether there are similar theories in adversarial learning. I think it will help readers to better understand them and it will not reduce your contributions.

**Summary Of The Paper:**

This paper proposes the first black-box hard-label backdoor detection where the defender can only access the predicted label of queries. Specifically, the authors first prove that the detection problem is bounded by an `adversarial objective’, based on which they reveal the adversarial singularity phenomenon that the perturbation generated by the adversarial objective is highly skewed distributed. Based on this phenomenon, the authors propose the adversarial extreme value analysis (AEVA) to detect backdoors in a black-box manner. The authors verify the effectiveness of their method in defending against BadNets, Blended Attack, Label-Consistent Attack, and Invisible Attack on CIFAR-10, CIFAR-100, and TinyImageNet dataset.

**Summary Of The Review:**

A practical and novel backdoor detection with theoretical guarantees.

---

### Official Review · Reviewer_9qnh · 2021-10-30

**Correctness:** 4
**Technical Novelty And Significance:** 4
**Empirical Novelty And Significance:** 4
**Recommendation:** 8
**Confidence:** 4

**Main Review:**

### Strengths and Weaknesses:

+ The proposed setting and algorithm are novel.
In particular, the subtle connection between backdoored networks and sparse adversarial example generation can inspire further research in this direction.

+ The analysis in the paper is logical, and it guides the reader well through the thought-process behind the algorithm.

+ The experimental results are comprehensive.
Although the experiments do not contain the most recent backdoor detection methods as baselines, this reviewer believes that the current comparison baselines (neural cleanse [[5](https://ieeexplore.ieee.org/abstract/document/8835365)] and DL-TND [[6](https://arxiv.org/abs/2007.15802)]) are enough to provide the bigger picture.
This is since almost all existing methods are typically designed for the white-box scenario, and as such, they have a huge advantage compared to the current method.

+ The paper is generally well-written, although some parts need proofreading.
See the minor comments about this below.

+ There are two gray areas regarding this submission that needs to be clarified further:

	1. Although the authors talk about the practicality of the black-box scenario in the introduction, this assumption needs to be justified further.
I strongly suggest providing a concrete example, where the use-case of this scenario is explained.
In this example, please specify the training data, the model, the user, and why the user who does not own anything from the training data to the model should be worried about backdoors?

	2. Another interesting question that is not been explored is the white-box performance of the algorithm.
Currently, the only part where the black-box scenario is being dealt with is the adversarial example generation.
Now, assuming that the user has access to the model parameters, what does the current approach provide in contrast to existing methods?
Asked the other way around, can someone use the Monte-Carlo gradient estimation in conjunction with existing methods to make them black-box?
Since being "black-box" is considered as the strength of the proposed method compared to existing ones, these questions need to be answered.

### Further Questions:

1. Can you clarify what this sentence below Eq. (3) means: "Additional difficulty comes from the fact that..."?

2. At the bottom of page 4 the paper reads: "Following this, the optimization in Eq. (3) converts to..." I am guessing that you meant only the first part of Eq. (3), right? Otherwise Eq. (4) is missing a $||m||$ term.

3. For the empirical study of Section 3.2, do you solve Eq. (6) in a white-box setting, or using the Monte-Carlo gradient estimator? How about Figure 6, is this figure generated in the black-box or white-box setting?

4. Does Eq. (7) mean that there are two ways to get the GAP: (1) by generating adversarial perturbations for multiple inputs (as the AVEA does) (2) by generating multiple perturbations for a single input?

5. Does the ablation study on "the impact of trigger size" mean that a scattered trigger is more likely to circumvent the detector than a sparse trigger?

6. The results reported in Appendix J indicate that in some cases the performance of the black-box detector is better than the white-box baselines, especially neural cleanse. What do you speculate is the reason behind this?

### Minor Comments/Suggestions:

+ Across the paper, the index "$i$" has been used to point to training samples (Eq. (2)), validation samples (Eq. (3)), and class (Algorithm 1).
Consider using a different index for each one of these to avoid misunderstanding.

+ Consider using different line styles and a bigger plot size for AUROC figures.

+ Consider adding a table of contents and/or explanation of different parts of the Appendix. Right now there are sudden jumps from one section to the next with no explanations.

+ In the second sentence after Eq. (15) $q$ has been used to point out the first and last dimensions of $X$. The second one needs to be changed to $p$.

+ Omit the $'$ in Eq. (16).

+ In Eq. (19) $N_b \to \infty$, not $N$.

+ In the explanations that follow Eq. (24), $K$ is used instead of $k$.

+ In Eq. (27) there is a $\hat{S}$ missing in the second $\log$.

+ Figure 13 is colliding with the text.

**Summary Of The Paper:**

This paper presents a novel approach for the detection of backdoored neural networks.

Inspired by the deployment of third-party networks in cloud services, it is argued that this detection task needs to be done in a black-box, hard-label scenario.
In this regime, it is assumed that the user/defender only has access to the model through making queries and getting back the labels for those queries.
To solve this newly proposed problem, first, a connection between the _detection of backdoors_ and _adversarial example generation with sparse perturbations_ is drawn.
Then, it is shown that for backdoor models solving this adversarial example generation objective leads to perturbations that are mostly concentrated in the backdoor mask area.
This argument is shown both theoretically (for linear classifiers) and empirically (for deep neural networks).

Based on this observation, which the paper calls the _adversarial singularity phenomenon_, a practical black-box backdoor detection is proposed.
Specifically, this algorithm first computes the aforementioned adversarial perturbations for some validation data in each class.
To this end, a Monte-Carlo gradient estimation is used to make the algorithm suitable for the black-box setting.
Then, based on the maximum value of these perturbations, it is decided whether the network for this label is infected or not.
The effectiveness of this method, named AVEA, is shown through extensive experiments and ablation studies on various datasets (CIFAR-10, CIFAR-100, and Tiny-ImageNet) and attacks (BadNets [[1](https://arxiv.org/abs/1708.06733)], Label-consistent [[2](https://arxiv.org/abs/1912.02771)], Watermark [[3](https://arxiv.org/abs/1712.05526)], and Invisible [[4](https://arxiv.org/abs/1909.02742)] attack).

**Summary Of The Review:**

While there are some gray areas around the proposed method (please see the strengths and weaknesses), I believe that this is a well-written, thought-provoking paper that can be interesting to the community and bring forward fruitful discussions.
As such, I vote for borderline acceptance of the paper.
If the authors can provide convincing answers to my two questions in the main review, I would be happy to increase my score.

---

### Official Review · Reviewer_Zz4A · 2021-11-03

**Correctness:** 3
**Technical Novelty And Significance:** 3
**Empirical Novelty And Significance:** 3
**Recommendation:** 6
**Confidence:** 4

**Main Review:**

Strengths:
1. The method assumes the black-box scenario, which is practical.
2. Some theoretical analysis is provided to establish the connection between adversary and backdoor attacks.
3. The idea is smart for using adversarial attack results to detect backdoors.

Weakness:
1. It seems that it is not difficult to avoid being detected if the backdoor patterns are smartly designed.
2. There should be some work using interpretation to detect the backdoor. Due to the close relation between adversarial attack and interpretation, I am not sure if the proposed method is still novel from this perspective.

**Summary Of The Paper:**

The paper propose a new backdoor detection method. The method does not require the original poisoned training data or the parameters
of the target DNNs. Given an image, the proposed method initiates adversarial attack on it. If the model contain backdoors for predicting the target label, then the adversarial noise will contain singular patterns, which could be detected using extreme value analysis.

**Summary Of The Review:**

The paper proposes a good idea for using adversarial attack patterns to diagnose if backdoors exist in models. My only concerns are twofold. First, some adaptive studies could be conducted, analyzing the scenarios where the proposed detection method could be circumvented. This may not be very difficult since this paper assumes that the backdoor patterns are focused patches. Second, the proposed idea seems to be similar to using interpretation (e.g., heat maps) to detect backdoor, since adversarial attack = inversed interpretation. From this perspective, the proposed idea does not look that novel.

---

### Official Review · Reviewer_TYjG · 2021-11-03

**Correctness:** 4
**Technical Novelty And Significance:** 2
**Empirical Novelty And Significance:** 2
**Recommendation:** 6
**Confidence:** 4

**Main Review:**

I have several concerns as follow:

- The relationship between adversarial examples and backdoor samples (e.g., pictures with auxiliary patches) is well known. Therefore, it is reasonable to expect that the adversarial singularity phenomenon may not occur in backdoor-infected DNNs. Will algorithm 1 be efficient under cases where the infected range and uninfected range overlap a lot and the threshold T is large (as discussed in Sec. 3.3)?

- It seems that the black-box setting is somehow contrived since it only poses challenges on computing the gradient ∇φ(x,yt). What is the effect of the sample size used for gradient estimation on the detection accuracy? With exact gradient computations, will the proposed algorithm 1 (AGAP) outperform the existing white-box detection methods?

- The result of Lemma 1 seems to be a natural consequence under the linear model assumption, mean-squared-error loss, and the optimization formulation of Eq. 6. Since this paper targets the adversarial extreme value analysis in DNN, do we have a more formal understanding of the regime of modern neural network architectures?


**Summary Of The Paper:**

This paper proposed an adversarial extreme value analysis (AEVA) framework to detect backdoors in black-box neural networks.
Specifically, they first obtained a new (upper bound) backdoor detection formulation by using convex relaxation. With linear model assumption and mean squared error loss, they showed that the mass in the adversarial perturbation would be occupied in the mask area as the backdoor sample size goes to infinity. Hence, a highly skewed distribution in the adversarial map for the infected label is expected.
Based on the above observation, they designed an Aggregated Global Adversarial Peak (GAP) for detecting the adversarial maps. With the limitation of the black-box setting, a Monte Carlo-based gradient estimation was used in the GAP.
Finally, they evaluated their methods on several real-world datasets.

**Summary Of The Review:**

Overall, I think the paper provides a new perspective of backdoor defense, but it could be made stronger by addressing some critical aspects as listed above.

---

### Author Response · Authors · 2021-11-16
**General Response**

We want to thank all reviewers for the detailed comments and constructive suggestions. We have updated a new version of our submission, which includes additional evaluation on dynamic and non-compact backdoor triggers (Appendix.K), detailed analysis on adaptive attacks, as well as improving the overall presentation of our paper according to the review comments.

---

### Public Comment · ~Junfeng_Guo2 · 2022-02-26
**Code for AEVA**

Code for **AEVA** can be found at **https://github.com/JunfengGo/AEVA-Blackbox-Backdoor-Detection-main**.
Hope our work can lead to addressing the **Backdoor Detection Problems** in the **Black-Box settings**!

---

### Decision · Program_Chairs · 2022-01-20

**Decision:**

Accept (Poster)

**Comment:**

This work proposed to detect backdoor in a black-box manner, where only the model output is accessible.

Most reviewers think it is a valuable task, and this work provides a novel perspective of using adversarial perturbation to diagnosis the backdoor. Some theoretical analysis for linear models and kernel models are provided. There is still huge gap to analyze the DNN model. But on the other side, it provides some insight to understand the proposed method and could inspire further studies.

Besides, since there have been many advanced backdoor attack methods, and many more are coming out, I am not sure that the proposed detection criteria is well generalizable, considering only some typical attack methods are tested. However, I think the studied problem is valuable, and the presented analysis is inspired for future works. Thus, I recommend for accept.